# Green Synthesis of Unsaturated Fatty Acid Mediated Magnetite Nanoparticles and Their Structural and Magnetic Studies

Amlan Kumar Das [1],*, Apoorva Fanan [2], Daoud Ali [3], Vijendra Singh Solanki [1],*, Brijesh Pare [4], Bader O. Almutairi [3], Neha Agrawal [5], Neera Yadav [6], Vikram Pareek [1] and Virendra Kumar Yadav [7],*

[1] Department of Chemistry, School of Liberal Arts and Sciences, Mody University of Science and Technology, Lakshmangarh, Sikar 332311, India
[2] Department of Nanoscience and Technology, School of Engineering and Technology, Mody University of Science and Technology, Lakshmangarh, Sikar 332311, India
[3] Department of Zoology, College of Science, King Saud University, P.O. Box 2455, Riyadh 11451, Saudi Arabia
[4] Department of Chemistry, Govt. Madhav Science, PG College, Vikram University, Ujjain 456001, India
[5] Department of Chemistry, Navyug Kanya Mahavidyalaya, University of Lucknow, Lucknow 226004, India
[6] College of Pharmacy, Gachon University, Incheon 21999, Republic of Korea
[7] Department of Biosciences, School of Liberal Arts and Sciences, Mody University of Science and Technology, Lakshmangarh, Sikar 332311, India
* Correspondence: amlan.snigdha72@gmail.com (A.K.D.); vijendrasingh0018@gmail.com (V.S.S.); yadava94@gmail.com (V.K.Y.)

**Abstract:** The green, cost-effective and sustainable synthesis of nanomaterials has been a key concern of scientists and researchers. In this view, MNPs were prepared using a sapota plant leaf extract and the surface of the magnetite nanoparticles was engineered with unsaturated fatty acids. The first report on the effect of unsaturation on the size and magnetic properties of magnetite nanoparticles (MNPs), prepared by the co-precipitation method, has been studied by coating surfactants on MNPs based on their unsaturation from zero to three (lauric acid, oleic acid, linoleic acid, linolenic acid). The size effect and magnetic properties of MNPs coated with a surfactant have been studied in comparison with uncoated magnetite nanoparticles. After the surface modification of the magnetite particle, it is necessary to check whether the magnetic property has been restored or not. Therefore, the magnetic property was studied. The presence of a surfactant on the surface of MNPs was confirmed by Fourier-transform infrared spectroscopy (FTIR), which was later confirmed by scanning electron microscope (SEM) and thermogravimetric analysis (TGA). The atomic structure was studied by X-ray diffraction (XRD) and the size of uncoated and surfactant-coated MNPs was determined by transmission electron microscopy (TEM) and the Scherrer equation by following XRD data. The magnetization property was analyzed by a vibrating sample magnetometer (VSM) at 10, 100 and 300 K and both bared and surfactant-coated MNPs exhibited a superparamagnetic nature at room temperature. The saturation magnetization ($M_s$) study shows that MNPs coated with a surfactant have a lower saturation magnetization value in comparison to uncoated NPs, confirming surface layering. Because the magnetic fluid has been stabilized in the aqueous medium, the double-layer model is expected to prevail.

**Keywords:** green synthesis; unsaturated fatty acids; surface engineering; bilayer structure; structural and magnetic properties

## 1. Introduction

The synthesis of nanomaterials via different methods and their utilization in various fields of science and technology has yielded a revolution in the industrial and manufacturing sectors. Due to their unique properties, such as electrical, magnetic [1,2], and optical properties, these metal oxide NPs are tremendously used in material science, engineering, and even in medical sciences [3].

MNPs (Fe$_3$O$_4$) show unique properties, such as being eco-friendly, stable, and biodegradable, having a high surface area, and being cost-effective and biocompatible in nature. These are used as a catalyst for the degradation of organic toxic pollutants from the environment and the separation of biomaterials [4–6].

Due to their higher hydrophilicity, MNPs have some limitations, such as being coagulated in an aqueous medium and oxidized in the air, as well as having instability in the acidic medium and less extraction capacity. In this view, to overcome the weaknesses of MNPs, their surface can be saturated and modified by monocarboxylic fatty acids, which are byproducts of the different vegetable oils and hydrocarbons available from the purifications of the oils [7].

Surfactants have unique properties to prevent MNPs from aggregating and they can be used to stabilize the magnetism of MNPs in aqueous and non-aqueous mediums by surface modification from the surfactants.

Coating with surfactants reduces NPs' agglomeration by forming a covering layer on the surface of MNPs. In that layer, a hydrophilic head will stick to the hydrophilic magnetite nanoparticles through covalently bonded chemical adsorption, leaving the tail to the aqueous medium, which eventually stabilizes by double-layer formation through physical adsorption, leaving the hydrophilic head of the carboxylate anion to face the aqueous medium. This results in the bilayered structure of the surfactant–magnetite nanoparticle hybrid [8].

In this research paper, four fatty acids (lauric acid or dodecanoic acid, oleic acid, linoleic acid and linolenic acid) have been used for the saturation of the MNPs surface. Figure 1 shows the MNPs coated with fatty acids.

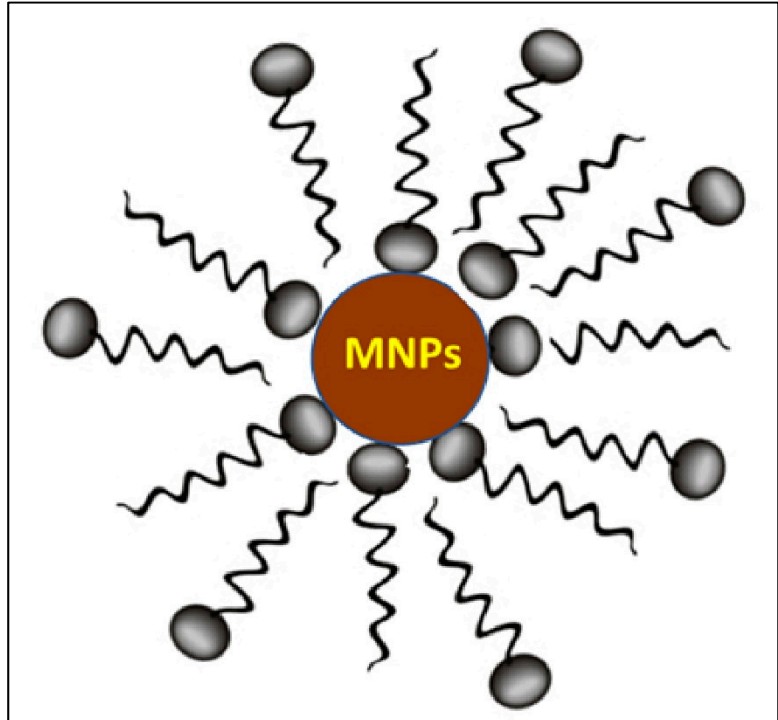

**Figure 1.** MNPs coated by fatty acid.

## 2. Materials and Methods

All the chemicals, including FeCl$_3$·6H$_2$O, FeSO$_4$·4H$_2$O, ethanol, ammonia and fatty acids for the synthesis of coated MNPs were purchased from Sigma Aldrich Chemical Pvt. Ltd. and Merck chemicals. (Sigma Aldrich Chemical Pvt. Ltd. and Merck chemicals., Bangalore, Karnataka, India). The standard solution was prepared with double-distilled water.

*Synthesis of MNPs*

Magnetite nanoparticles ($Fe_3O_4$) were synthesized by following the co-precipitation method from the sapota plant leaf extract [9–12]. MNPs were coated with an unsaturated fatty acid, forming a double layer and giving rise to colloidal and magnetic stability to the magnetite nanoparticle. Sapota plant leaf was collected from the university campus, dried and boiled with 100 mL of distilled water for 1 h. The aqueous solution was filtered, and the extracts were used for further synthesis of the magnetite nanoparticles. The ferrous and ferric chloride solution in a molar ratio of 1:2 was dissolved in distilled water and 15 mL of sapota plant leaf extract under magnetic stirring for 15 min, followed by a drop-by-drop addition of the surfactant solution made by dissolving 60 mM lauric/oleic/linoleic/linolenic acid in distilled water/ethanol. After the complete mixing of the above chemicals, pure ammonia was added as a precipitant at 80 °C for 15 min, followed by magnetic stirring at 80 °C for 1 h. A black precipitate solution formed, which was centrifuged at 10,000 rpm for 10 min, followed by washing twice with distilled water. The obtained black precipitate was dried in a hot air oven for 4 h at 100 °C.

## 3. Characterization of MNPs

The initial confirmation of the presence of unsaturated fatty acids on the surface of MNPs was conducted by FTIR spectroscopy. It was used to determine the functional group of active compounds based on the peak value in the infrared region. Another confirmation for the presence of a surfactant and the morphology of surfactant-coated MNPs was examined by using a scanning electron microscope (SEM). The presence of a surfactant on the surface of MNPs was analyzed in terms of a change in the physical and chemical properties of materials with an increasing temperature or as a function of time by a thermogravimetric analyzer. The crystalline structure of the sample was identified by X-ray diffraction. The X-rays were produced using a sealed tube and the wavelength of the X-ray was 1.54 nm Cu K-alpha radiations. The diffraction patterns were carried out using a 2θ range of 25–70. The particle size, lattice parameter and area were calculated by applying the Scherrer equation to the obtained XRD data. The exact size of the uncoated and surfactant-coated MNPs was determined by a transmission electron microscope. Magnetization was measured by means of a 14 T PPMS vibrating sample magnetometer (VSM). Zero field cooling (ZFC) and field cooling (FC) was performed by cooling the sample to 300 K, and in the presence of the external field of 500 Oe and $M/M_s$, it was studied at 10, 100 and 300 K.

### 3.1. FTIR Analysis

FTIR spectroscopy was used to identify the functional groups of the active components based on the bands in the region of infrared radiation. In Figure 2, the FTIR spectra showed a band at ~3452 $cm^{-1}$, which is assigned for the stretching vibration of –OH, i.e., absorbed by $Fe_3O_4$ NPs. The bands at 2951 $cm^{-1}$ and 2850 $cm^{-1}$ are assigned for stretching vibrations in the methylene (H=C=H) group of arachidonic acid. The bands at 1626 $cm^{-1}$ and 1408 $cm^{-1}$ are assigned for the stretching vibration of C=C, and the band at 1574 $cm^{-1}$ is assigned for the vibration in C=O of arachidonic acid, whereas the band at 577 $cm^{-1}$ shows the stretching vibration of Fe-O of the surfactant-coated MNPs [13–15]. FTIR spectra of pure $Fe_3O_4$ show a band at 3452 $cm^{-1}$ assigned for –OH and 573 $cm^{-1}$, which shows stretching vibration in Fe-O of pure MNPs [16–19]. Yadav et al. 2020 also obtained bands for Fe-O in the range of 400–800 $cm^{-1}$ [15].

### 3.2. TGA Analysis

TGA results in Figure 3a show that MNPs coated with one of the surfactants oleic acids had an initial weight loss of 1.44% at around 150 °C, followed by a gradual loss of 3.84, 13.42 and 8.20% at around 270, 445 and 622 °C, respectively. At 699.5 °C, the residual mass was only 70.57% but, in comparison to the coated MNPs, uncoated MNPs had a 97.10% residual mass at 699.5 °C shown in Figure 3b [20,21].

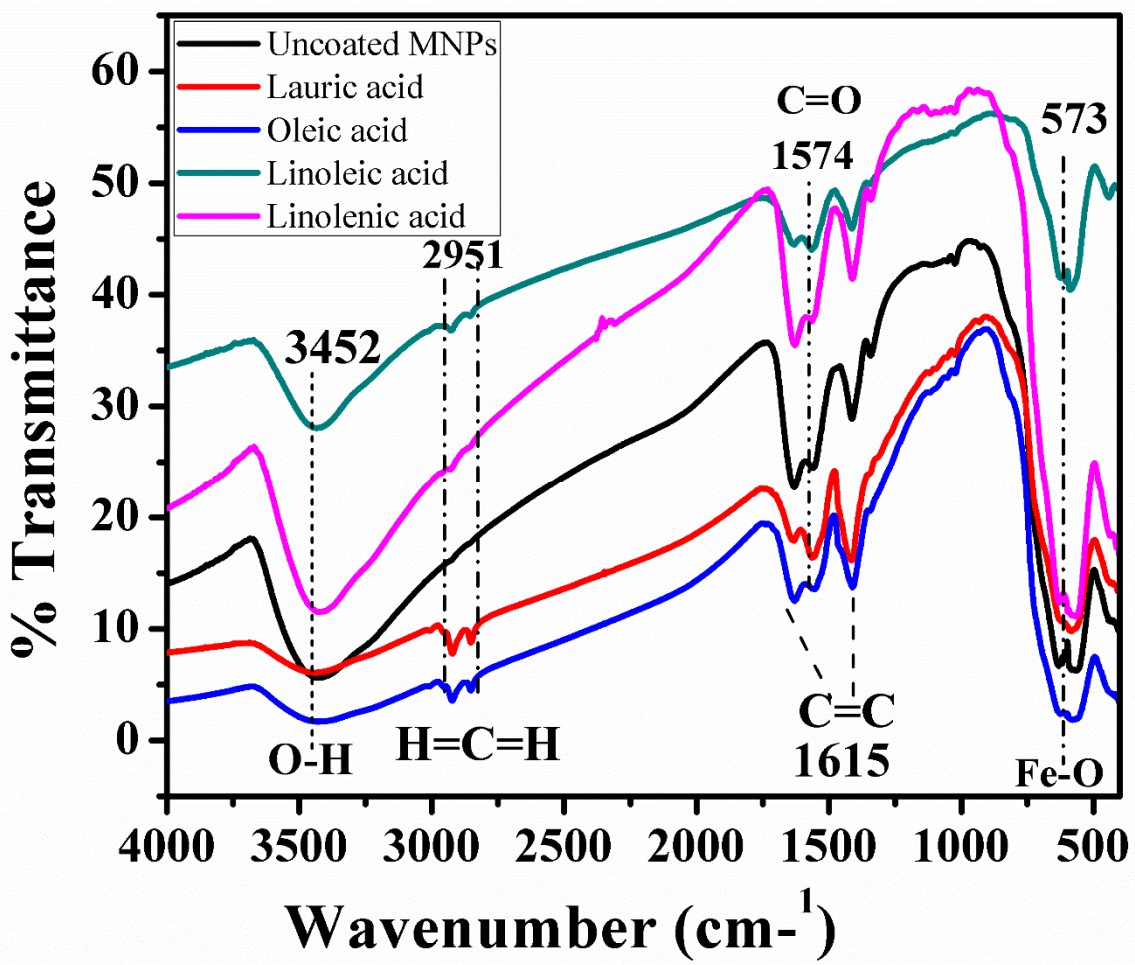

**Figure 2.** TIR spectra of uncoated and surfactant-coated MNPs.

### 3.3. SEM Analysis

A scanning electron microscope (SEM) of uncoated MNPs and all the fatty-acid-coated MNPs are shown in Figure 4a–f. Figure 4a,b shows the SEM micrographs of uncoated MNPs, which are highly aggregated. Figure 4c shows a SEM micrograph of lauric-acid-coated MNPs, Figure 4d shows oleic acid-coated MNPs, Figure 4e shows a SEM micrograph of linoleic-acid-coated MNPs, and Figure 4d shows a SEM micrograph of linolenic-acid-coated MNPs. The size of all the particles is 10–30 nm, but due to aggregation and their very small size, they exhibit large lumps. The particles are highly aggregated as shown in the images. Several investigators have also reported a similar morphology of the synthesized MNPs, such as Yadav and Fulekar (2018), [22] and Yadav et al. 2020 [15]. The authors have obtained a range of sizes, i.e., 20 nm to 120 nm, by the chemical co-precipitation method [22]. Besides this, Pandya et al. 2018 also obtained similar results for synthesized MNPs using the chemical co-precipitation method. The particle size distribution was around 20–100 nm. It was polydispersed in nature, and as the unsaturation increased, particle size decreased.

### 3.4. XRD Analysis

The black precipitate was formed by the chemical co-precipitation method, and it was identified as magnetite by the X-ray diffraction pattern. The diffraction pattern for bared and surfactant-coated magnetite is shown in Figure 5. All the peaks were indexed as magnetite (reference code: JCPDS 01-088-0315), indicating the presence of a single-phase cubic spinel structure and the diffraction peaks were fairly broad. The maximum intensity

peak was found at 35.3–35.5⊖ (311) [23], which indicates the formation of the magnetite phase of iron oxide nanoparticles.

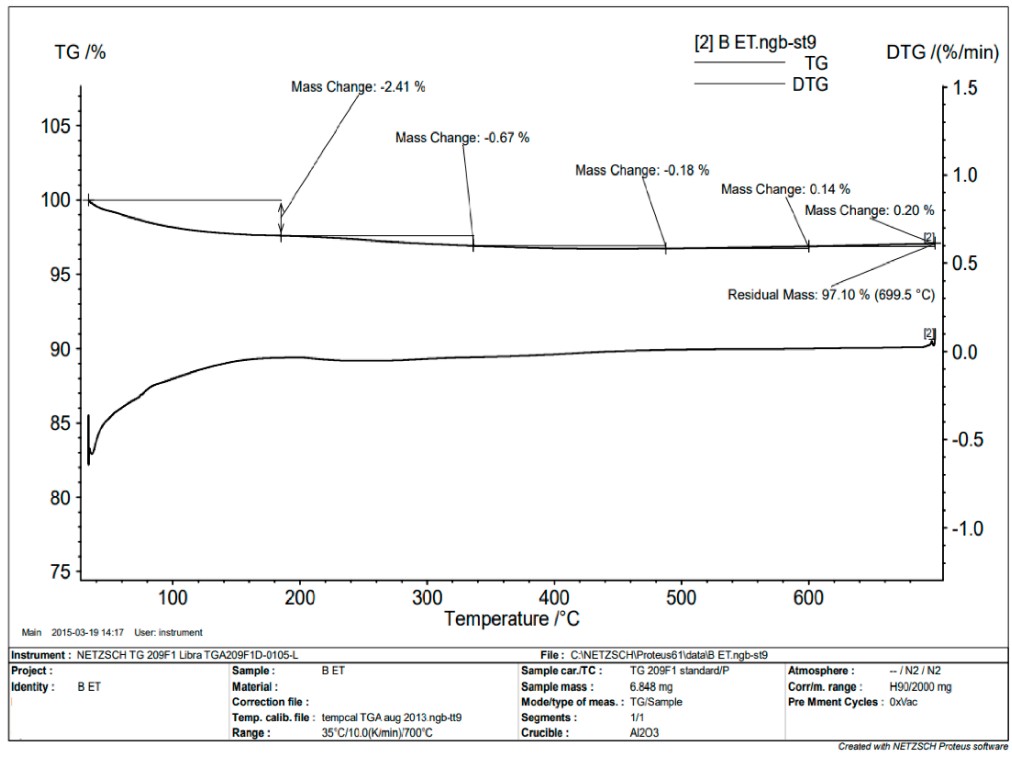

(**a**)

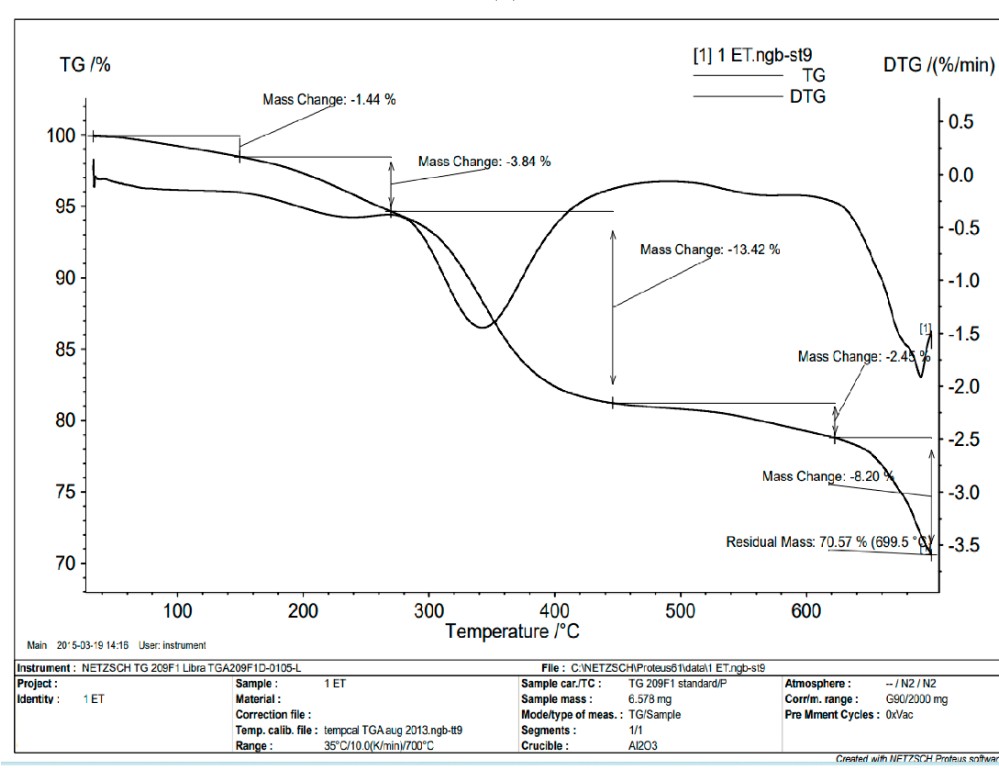

(**b**)

**Figure 3.** GA analysis for (**a**) surfactant-coated MNPs, (**b**) uncoated MNPs.

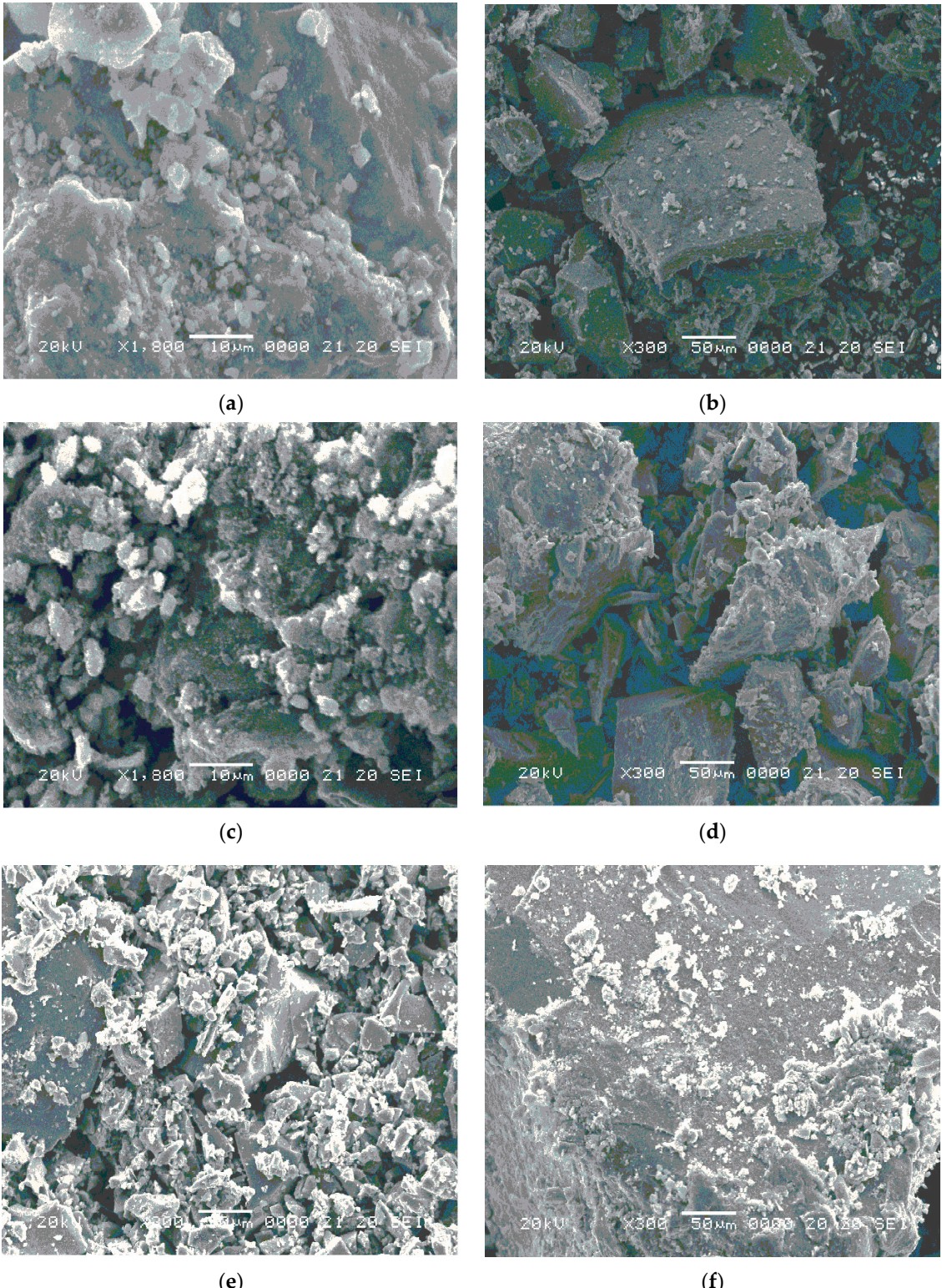

**Figure 4.** EM images of (**a**,**b**) uncoated MNPs, (**c**) lauric-acid-coated MNPs, (**d**) oleic-acid-coated MNPs, (**e**) linoleic-acid-coated MNPs, (**f**) linolenic-acid-coated MNPs.

The size, shape, lattice parameter and area of bared MNPs and surfactant-coated MNPs were analyzed using XRD data by following the Scherrer equation [24].

$$\tau = K\lambda/\omega \, cos\theta$$

where $\tau$ is the mean size of the ordered domain, $K$ is the dimensionless shape factor, which is 0.9, $\lambda$ is the X-ray wavelength, $\omega$ is the width on the $2\theta$ scale and $\theta$ is the Bragg angle. The obtained results are shown in Table 1.

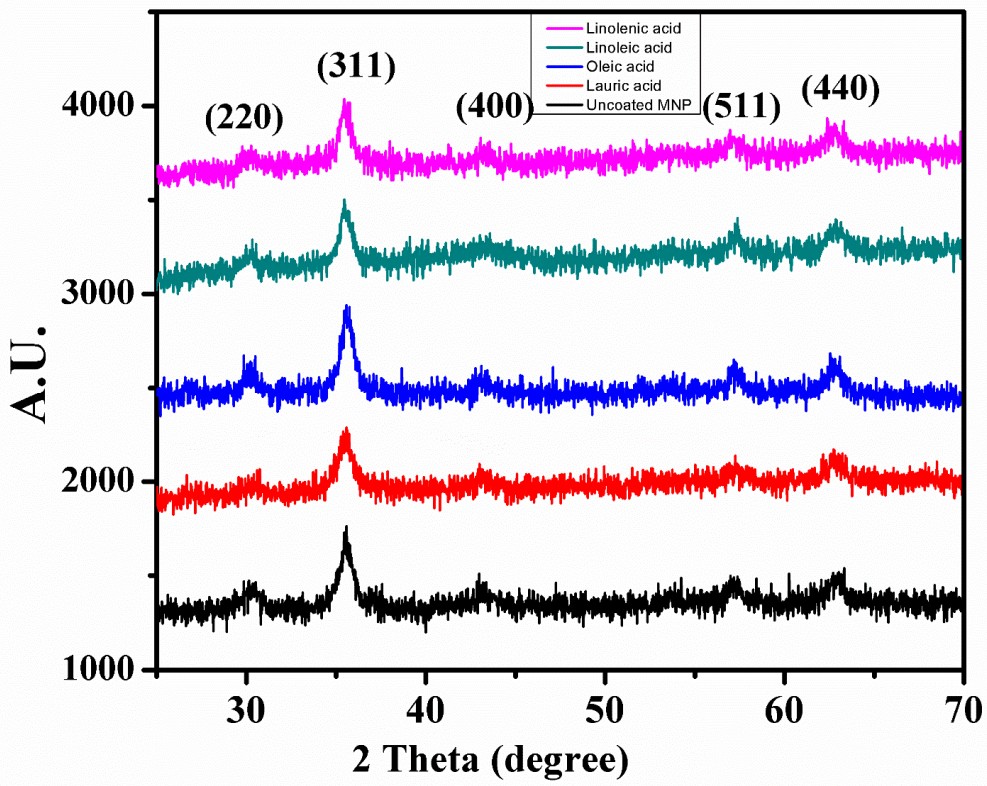

**Figure 5.** X-ray diffraction pattern of uncoated MNPs and surfactant-coated MNPs.

**Table 1.** Results obtained from XRD analysis.

| Sr. No. | MNPs Coated by | Shape | Area | Lattice Parameter | Particle Size (nm) | |
|---|---|---|---|---|---|---|
| | | | | | By XRD | By TEM |
| 1. | Uncoated | Cubic | 263.72 | $8.36577 \times 10^{-10}$ | 15 | ~16 |
| 2. | Lauric acid | Cubic | 193.32 | $8.35144 \times 10^{-10}$ | 12 | ~14 |
| 3. | Oleic acid | Cubic | 307.80 | $8.37072 \times 10^{-10}$ | 10 | ~12 |
| 4. | Linoleic acid | Cubic | 221.86 | $8.3763 \times 10^{-10}$ | 8 | ~10 |
| 5. | Linolenic acid | Cubic | 207.76 | $8.3763 \times 10^{-10}$ | 6 | ~8 |

The obtained result is in agreement with previous results obtained for MNPs. Yadav and Fulekar (2018) and Yadav et al. 2020 also obtained peaks at 33 and 35°, which was attributed to the hematite and magnetite phase of the IONPs [22,25–28].

### 3.5. TEM Analysis

The effect of coating with unsaturated fatty acids will give magnetite nanoparticles colloidal stability and superparamagnetic stability by forming a double layer around the magnetite nanoparticle. Figure 6a,d shows TEM images of uncoated MNPs. The particle is of a cuboidal to spherical shape, whose size varies from 7–10 nm. This small size of the MNPs indicates the formation of the superparamagnetic nanoparticle. Figure 6e,f shows TEM images of lauric-acid-coated MNPs. The size of the particle is 4–8 nm as revealed by TEM, whereas the particle is spherical to cuboidal in shape. Figure 6g,h shows TEM images of oleic-acid-coated MNPs whose size is 4–8 nm and whose shape is also spherical

to cuboidal. Figure 6i,j shows TEM images at 20 nm where the size of linoleic-acid-coated MNPs varies from 5–15 nm and the shape is cuboidal to spherical. The linolenic-acid-coated MNPs are shown in Figure 6k,l, whose size varies from 5–14 nm. Here, the morphology is also mainly spherical to cuboidal in shape. So, in all the types of MNPs, the size varies from ~16 nm to ~8 nm as shown in Figure 6a–k. All the particles showed aggregation, and this could be due to their very small size. In all cases, the average particle size of the synthesized magnetite nanoparticle hybrid was calculated using the Debye–Scherrer formula ranging from 12 nm to 6 nm, which is much less than what was obtained from TEM measurements for the same surfactant hybrid samples and confirms the presence of the non-crystalline surfactant layer on the surface of the magnetite nanoparticles [29]. Yadav and Fulekar (2018) [22], and Yadav et al. 2020 [15] reported the synthesis of the cuboidal to spherical shape of the MNPs by the chemical co-precipitation method. The size of uncapped MNPs varied from 8–20 nm.

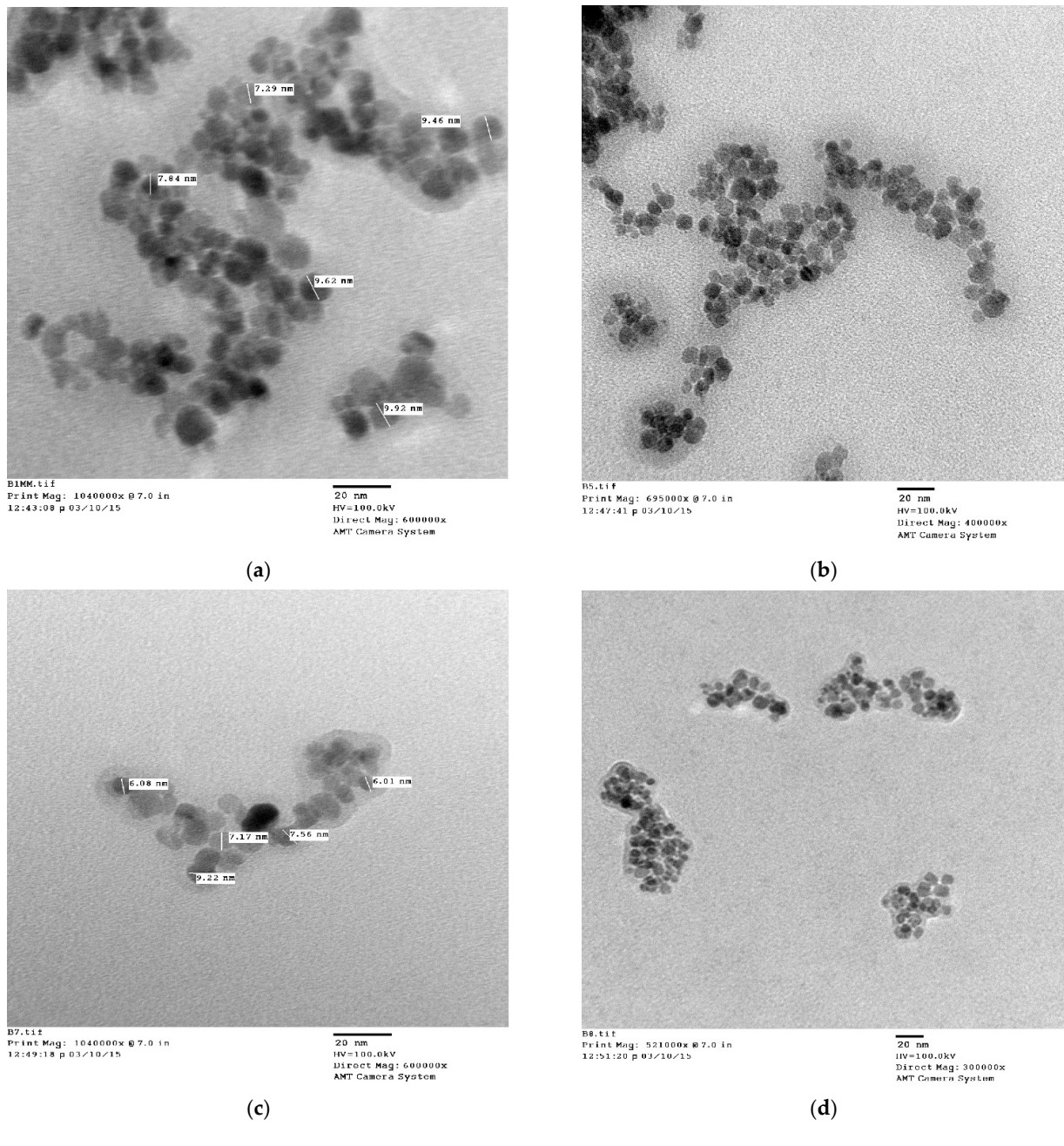

**Figure 6.** *Cont.*

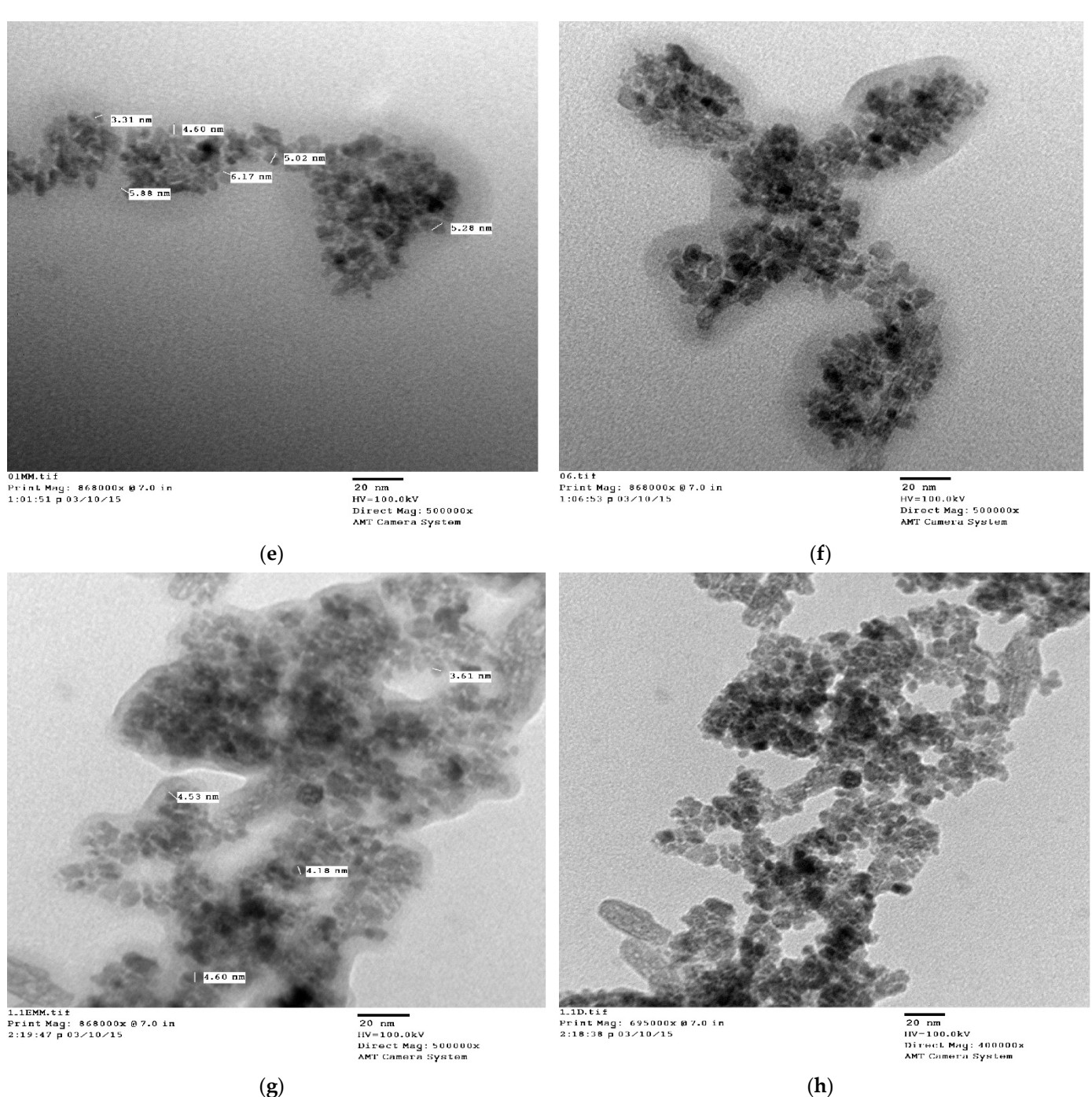

**Figure 6.** *Cont.*

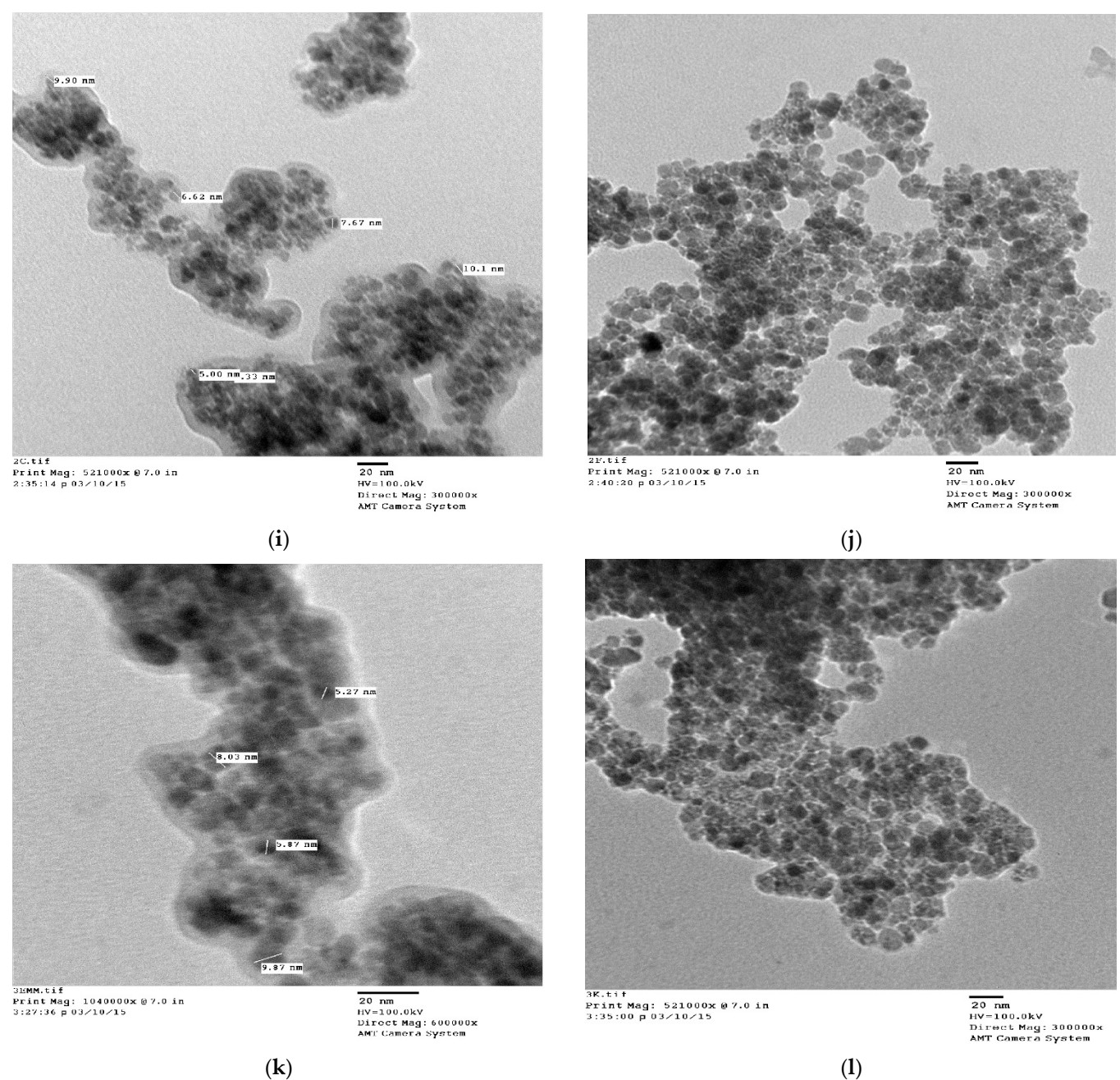

**Figure 6.** TEM images of (**a–d**) uncoated MNPs, (**e**,**f**) lauric-acid-coated MNPs, (**g**,**h**) oleic-acid-coated MNPs, (**i**,**j**) linoleic-acid-coated MNPs, (**k**,**l**) linolenic-acid-coated MNPs.

## 4. Result and Discussion

*Magnetic Study*

MNPs at room temperature show superparamagnetic behavior, i.e., no hysteresis loop. However, a hysteresis loop can be achieved by pinning magnetic domain walls at grain boundaries within the material and by that material becoming ferromagnetic [30]. If synthesized NPs are of a size below a superparamagnetic critical size. In that case, hysteresis behavior vanishes above the blocking temperature. The magnetic behavior of MNPs was studied using VSM, ZFC and FC curves in the presence of the external field of 500 Oe. Increasing the temperature up to 300 K, as presented in Figure 7a–e, shows that these uncoated MNPs and surfactant-coated MNPs exhibit a superparamagnetic nature [31–34]

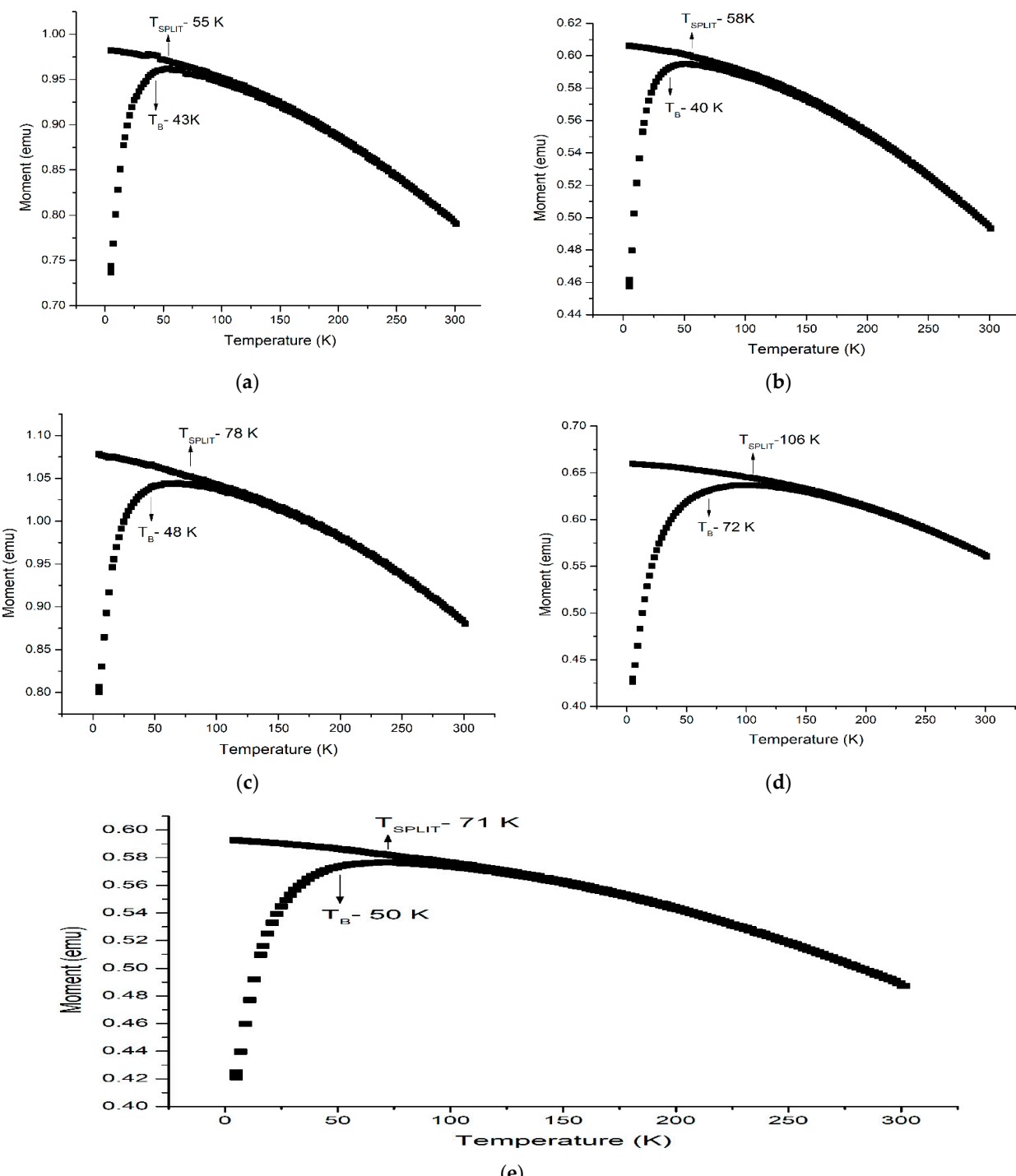

**Figure 7.** ZFC and FC curve for (**a**) uncoated MNPs, (**b**) lauric-acid-coated MNPs, (**c**) oleic-acid-coated MNPs, (**d**) linoleic-acid-coated MNPs, (**e**) linolenic-acid-coated MNPs.

The increase in blocking temperature with an increase in the unsaturation of the surfactant may be understood as the particle size decreasing the thickness of the double layer and subsequently increasing, making the effective size of the surfactant–magnetite hybrid bigger. This may indicate that the average distance between magnetite nanoparticles does not change significantly. It seems that the double layering took place on the surface of some already agglomerated magnetite nanoparticles and not on all the individual magnetite nanoparticles. From Figure 7a, we can see the loss of magnetization with the increase in temperature over 200 K is very sharp for uncoated MNPs in comparison with the bilayer-

coated MNPs, which are very smooth and slow. It can be understood that the bilayer of the surface of the magnetite acts as a shielding layer for the heat transfer to the inner magnetite, therefore reducing the rate of decreased magnetization.

The hysteresis magnetic loop M(H) was measured at 10, 100 and 300 K for both uncoated and surfactant-coated MNPs, presented in Figure 8. The saturation magnetization ($M_S$) of surfactant-coated MNPs, in comparison to bared MNPs, was observed at 10, 100 and 300 K, and it was observed that surfactant-coated MNPs have a lower $M_S$ value than uncoated MNPs. By comparing $M_S$ for surfactant-coated MNPs at 10, 100 and 300 K, it is found that oleic acid with a minimum Ms value will have the maximum extent of the adsorbed layer of a surfactant.

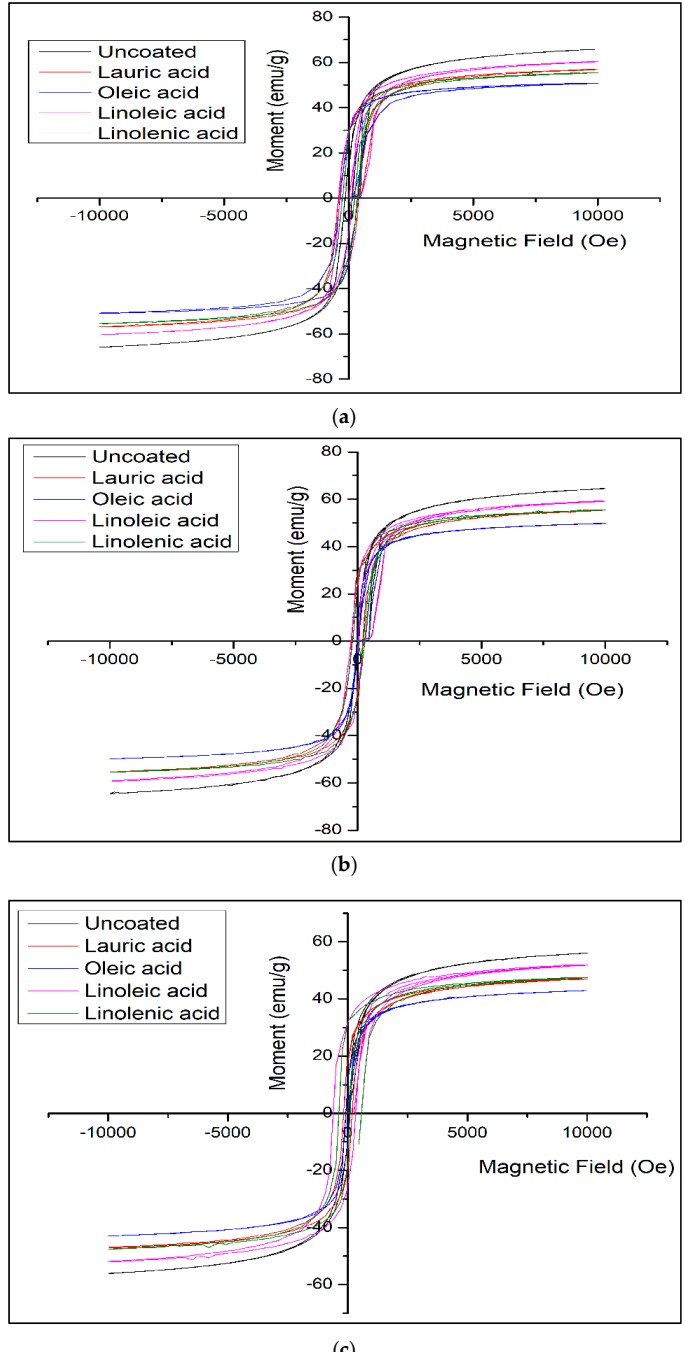

**Figure 8.** Comparison of $M_H$ curve for uncoated MNPs with surfactant-coated MNPs at (**a**) 10 K, (**b**) 100 K, (**c**) 300 K.

Surfactants with a higher unsaturation bring more fluidity to the double layer as the area per molecule adsorption increases with unsaturation because of a kink in their structure, which certainly makes the surfactant hybrid magnetite more biocompatible and less toxic. The extent of adsorption for higher unsaturated fatty acids is less confirmed by their saturation magnetization value, which is very much essential for these fatty acid magnetite hybrids to be influenced efficiently under an external magnetic field for their application in the biomedical field. Yadav et al. 2020 have also obtained similar results for the magnetic strength of the IONPs [35–37].

The variation of magnetic properties of all surfactant hybrids in comparison with uncoated MNPs has been tabulated below in Table 2.

**Table 2.** Variation of magnetic properties of surfactant hybrids in comparison to uncoated MNP.

| MNPs Coated by | $M_S$ (emu/g) | | | $H_C$ (Oe) | | | $M_R$ (emu/g) | | | $T_B$ (K) | $T_P$ (K) |
|---|---|---|---|---|---|---|---|---|---|---|---|
| | 10 K | 100 K | 300 K | 10 K | 100 K | 300 K | 10 K | 100 K | 300 K | | |
| Uncoated | 66 | 65 | 56 | 145 | 36 | 0 | 15 | 7 | 0 | 43 | 55 |
| Lauric acid | 57 | 55 | 47 | 152 | 100 | 0 | 16 | 10 | 0 | 40 | 58 |
| Oleic acid | 51 | 49 | 37 | 152 | 73 | 0 | 14 | 3 | 0 | 48 | 78 |
| Linoleic acid | 52 | 50 | 44 | 214 | 39 | 0 | 16 | 4 | 0 | 72 | 106 |
| Linolenic acid | 51 | 49 | 41 | 161 | 0 | 0 | 13 | 0 | 0 | 50 | 71 |

## 5. Conclusions

The present study shows a surface modification of MNPs using a range of unsaturated fatty acids. The synthesized MNPs and analysis by analytical instruments revealed their detailed information. The microscopic analysis, i.e., SEM and TEM, showed that particles are highly aggregated, with individual sizes varying from 14 nm to 8 nm. The band obtained in the region of 400–600 in FTIR confirmed the formation of MNPs due to the Fe-O bond. Additionally, the presence of various functional groups indicated the capping by unsaturated acids. Magnetic study shows that surfactant-coated MNPs have an increase in blocking temperature and splitting temperature in comparison to uncoated MNPs. The increase in blocking temperature with an increase in the unsaturation of the surfactant may be understood as the particle size decreases, the thickness of the double layer will also increase, making the effective size of the surfactant–magnetite hybrid bigger. The surfactant-coated MNPs with a lower saturation magnetization value than the uncoated MNPs at 10 K, 100 K and 300 K confirm the double-layer coating on the magnetite surface. Surfactants with more unsaturation in their tail will bring more fluidity to the double layer of the surface of the magnetite, making it more biocompatible and less toxic. Because of the presence of a double bond in the fatty acid tail, which brings polarity to the tail, the double layer eventually becomes more hydrophilic and the extent of water content increases; therefore, the water-soluble drug can easily be entrapped in the double layer. Therefore, this bilayer surfactant–magnetite hybrid could be the best to be used in targeted drug delivery.

**Author Contributions:** Conceptualization, A.K.D., A.F., B.O.A., N.A. and D.A.; data curation, N.A., N.Y. and V.K.Y.; methodology, V.S.S., N.A., A.K.D., V.P. and D.A.; validation, V.S.S., V.K.Y. and A.K.D.; formal analysis, V.K.Y., V.S.S. and D.A.; resources, N.Y., N.A. and A.F.; writing—original draft preparation, A.K.D., V.S.S., V.P., A.F., D.A. and N.A.; writing—review and editing, N.Y., B.O.A., V.S.S., V.P., A.K.D., V.K.Y. and A.F.; supervision, A.K.D., N.Y., V.S.S. and D.A.; project administration, V.S.S., A.K.D., V.P., V.K.Y. and N.A.; funding acquisition, B.O.A., D.A., N.Y., V.K.Y. and N.A.; investigation, B.P., V.P., A.F., V.S.S., N.A. and V.P.; software, B.O.A., V.K.Y., N.Y., V.P. and V.S.S.; visualization, A.K.D., D.A. and N.Y. All authors have read and agreed to the published version of the manuscript.

**Funding:** This research received no external funding.

**Institutional Review Board Statement:** Not applicable.

**Informed Consent Statement:** Not applicable.

**Data Availability Statement:** Not applicable.

**Acknowledgments:** This work was funded by the Researchers Supporting Project number (RSP-2021/165), King Saud University, Riyadh, Saudi Arabia AKD, AP, VKY and VSS are thankful to the Department of Chemistry, SLAS, and Mody University of Science and Technology for providing laboratory and instrument facilities for the research work.

**Conflicts of Interest:** The authors declare no conflict of interest.

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
