# Peer review of "Green Synthesis of Unsaturated Fatty Acid Mediated Magnetite Nanoparticles and Their Structural and Magnetic Studies"

_magnetochemistry, doi:10.3390/magnetochemistry8120174_

Round 1

Reviewer 1 Report

Reviewer general comment:

The authors have Designed and executed the experiment very well. In fact, the manuscript titled Green synthesis of Unsaturated Fatty acid Mediated Magnetite Nanoparticles and its Structural & Magnetic Studies is also written in a good way.

I have few reservations regarding the manuscript. Here are few

Reviewer comment 1:

English language must be revised in the manuscript

Reviewer comment 2:

In first paragraph of the introduction part, the key issue such as why the authors used specially sapota plant leaf extract or what are the benefits of using plant to prepare magnetite nanoparticles should be addressed.

Reviewer comment 3:

- Line 78: The authors should describe how they prepare the sapota plant leaf extract.

Reviewer comment 4:

Some grammatical mistakes and space problems exist throughout the manuscript. Authors should carefully revise the whole manuscript and do corrections accordingly.

Reviewer comment 5:

 Fig. 4 a , c & d are nor clear. Provide a good resolution picture.

Reviewer comment 6:

- Line 160: Deep discussion about effect of coating using unsaturated Fatty acid should be added.

Author Response

English language must be revised in the manuscript

Response:

Dear reviewer thank you for the comment, now we have corrected all the English language.

Reviewer comment 2:

In first paragraph of the introduction part, the key issue such as why the authors used specially sapota plant leaf extract or what are the benefits of using plant to prepare magnetite nanoparticles should be addressed.

Response:

Dear reviewer thank you very much for your valuable comments and suggestions, we have prepared magnetite nanoparticles by different leaf extract but in this case, we obtained the maximum yield of nanoparticles. Now a days various nanoparticles are synthesized by different leaf extract therefore we tried to synthesize magnetite nanoparticles by sapota plant extract.

Reviewer comment 3:

- Line 78: The authors should describe how they prepare the sapota plant leaf extract.

Response:

Dear esteemed reviewer, thank you very much for the comment, now we have incorporated method of preparation of leaf extract in the manuscript.

Reviewer comment 4:

Some grammatical mistakes and space problems exist throughout the manuscript. Authors should carefully revise the whole manuscript and do corrections accordingly.

Response:

Dear esteemed reviewer, thank you very much for the comment, now we corrected grammatical mistakes and space in the manuscript.

Reviewer comment 5:

 Fig. 4 a , c & d are nor clear. Provide a good resolution picture.

Response:

Dear esteemed reviewer, thank you very much for the valuable comment, now we have changed all the pictures to high resolution.

Reviewer comment 6:

Line 160: Deep discussion about effect of coating using unsaturated Fatty acid should be added.

Response:

Dear reviewer thank you very much for the valuable comment. The effect of coating by unsaturated fatty acid will give magnetite nanoparticles colloidal stability and also super para magnetic stability by forming double layer around the magnetite nanoparticle. Now this statement has been incorporated in the manuscript.

Reviewer 2 Report

Dear authors,

First, my congratulations for the effort put into the research and the manuscript.

However, the manuscript can be highly improved following some basic ideas:

1.     Extensive editing of English language and style must be performed to clarify the text and also the basic ideas you want to transmit to the reader, which are not perfectly clear.

2.     Clarifying the abstract: why magnetic measurements are performed?

3.     Clarifying the introduction and objectives. Last paragraph is not enough to explain all your intentions. I guess you should somehow mention you propose the coating of MNPs surface with fatty acids in order to prevent their aggregation, to improve their biocompatibility, to reduce their toxicity, and also to raise their hydrophilicity, enabling their use in biomedical applications such as drug delivery, being able to exploit a remarkable differential feature: their magnetism. Besides, the part explaining the bilayer should be better explained, indeed I didn’t understand what they claimed. The drawing shows a monolayer of fatty acids over MNPs, but the text says there is a bilayer. Besides the text says the hydrophilic head of the fatty acids will stick to the MNP, which is hydrophilic, but the drawing shows the opposite… Besides the text says the interaction between the hydrophilic head of the fatty acids and the MNP is a covalently bonded chemical adsorption, which is not correct. It may be covalently bonded, or chemically adsorbed, but not “covalently bonded chemical adsorption”. Again, this should be clarified, or at least better explained, even if there is no clue to guess if both phenomena take place, or there is a prevalence for one of them. And most important thing lacking in the intro. Why conducting a magnetic study of those coated or uncoated MNPs? Do the authors want to probe the coating? Do they want to probe the effect of unsaturation of fatty acids over magnetic properties? This must be explained.

4.     Improving materials and methods. The manuscript lacks a subsection explaining how MNPs were coated by surfactants.

5.     Improving characterization section. First paragraph should be included within previous section, explaining each technique (not including results), including magnetic measurements. In characterization section, magnetic results should also be included, and then proceed to the discussion. Therefore, maybe characterization should be included within results and discussion… ALL the images display indecent resolution. No letters / numbers can be read, therefore they are useless. Please provide images with enough resolution so that the reader can appreciate the data, and not imagine them. XRD letters and numbers can also be improved. TEM information within the images might be removed and changed by a simple scale bar.

1.     Concerning SEM images, the authors say “The presence of surfactant on the surface of MNPs was confirmed by Scanning Electron Microscope (SEM) shown in Figure 4”, but I do not understand how this is possible. The authors should explain and justify this, otherwise they should remove it. It is quite difficult for a scanning electro microscope to reveal organic and not conductive materials, which usually appear diffusive. I do not appreciate the fatty acids over MNPs, but if the authors can explain us, I would be willing to learn how to distinguish it from the inorganic part of the sample. Maybe if they had presented EDX results, they could have demonstrated the presence of organic matter (carbon) over the MNP. But there is no reference to this.

6.     Result and discussion. This section should not make reference to magnetic study, because there is no other subsection… Besides a great need of rephrasing the whole text, there is a continued lack of references when discussing key questions. For the sake of a better understanding, I will use an example: in lines 176-186, besides the need of rephrasing, the authors claim something which is not evident for all readers, so one or more references in different claims should be used to justify their reasonings. Same stands for the rest of the section.

7.     Conclusion. Some rephrasing should be performed in order to improve this key section.

In conclusion, there is a lot of effort to be put into this manuscript in order to improve it significantly. However, the experimental part seems to be correct, therefore I recommend accepting the manuscript after a minor revision (though the text editing should be quite extensive).

Author Response

First, my congratulations for the effort put into the research and the manuscript.

However, the manuscript can be highly improved following some basic ideas:

  1. Extensive editing of English language and style must be performed to clarify the text and also the basic ideas you want to transmit to the reader, which are not perfectly clear.

Response: Dear Reviewer thank you very much for the valuable comment, now we have corrected the English language in the manuscript.

  1. Clarifying the abstract: why magnetic measurements are performed?

Response: Dear Reviewer thank you very much for the valuable comment, after surface modification of the magnetite particle it is necessary to check whether the magnetic property has been restored or not. Therefore, magnetic measurements are performed.

  1. Clarifying the introduction and objectives. Last paragraph is not enough to explain all your intentions. I guess you should somehow mention you propose the coating of MNPs surface with fatty acids in order to prevent their aggregation, to improve their biocompatibility, to reduce their toxicity, and also to raise their hydrophilicity, enabling their use in biomedical applications such as drug delivery, being able to exploit a remarkable differential feature: their magnetism. Besides, the part explaining the bilayer should be better explained, indeed I didn’t understand what they claimed. The drawing shows a monolayer of fatty acids over MNPs, but the text says there is a bilayer. Besides the text says the hydrophilic head of the fatty acids will stick to the MNP, which is hydrophilic, but the drawing shows the opposite… Besides the text says the interaction between the hydrophilic head of the fatty acids and the MNP is a covalently bonded chemical adsorption, which is not correct. It may be covalently bonded, or chemically adsorbed, but not “covalently bonded chemical adsorption”. Again, this should be clarified, or at least better explained, even if there is no clue to guess if both phenomena take place, or there is a prevalence for one of them. And most important thing lacking in the intro. Why conducting a magnetic study of those coated or uncoated MNPs? Do the authors want to probe the coating? Do they want to probe the effect of unsaturation of fatty acids over magnetic properties? This must be explained.

Response: Dear reviewer thank you very much for the rigorous review and comment, our intention is to study the effect of fatty acid on the magnetic property of the magnetite nanoparticle. The diagram has been redrawn with the hydrophilic part of the fatty acid is attached to hydrophilic magnetite nanoparticles and double layer has been shown.

  1. Improving materials and methods. The manuscript lacks a subsection explaining how MNPs were coated by surfactants.

Response: Dear esteemed reviewer thank you for the valuable comment, and now we have improved in the manuscript. MNP were coated with unsaturated fatty acid forming a double layer giving rise to colloidal and magnetic stability to the magnetite nanoparticle.

  1. Improving characterization section. First paragraph should be included within previous section, explaining each technique (not including results), including magnetic measurements. In characterization section, magnetic results should also be included, and then proceed to the discussion. Therefore, maybe characterization should be included within results and discussion… ALL the images display indecent resolution. No letters / numbers can be read, therefore they are useless. Please provide images with enough resolution so that the reader can appreciate the data, and not imagine them. XRD letters and numbers can also be improved. TEM information within the images might be removed and changed by a simple scale bar.

Response: Dear reviewer thank you very much for the valuable comment, now we have added all the images with high resolution in the manuscript.

  1. Concerning SEM images, the authors say “The presence of surfactant on the surface of MNPs was confirmed by Scanning Electron Microscope (SEM) shown in Figure 4”, but I do not understand how this is possible. The authors should explain and justify this, otherwise they should remove it. It is quite difficult for a scanning electro microscope to reveal organic and not conductive materials, which usually appear diffusive. I do not appreciate the fatty acids over MNPs, but if the authors can explain us, I would be willing to learn how to distinguish it from the inorganic part of the sample. Maybe if they had presented EDX results, they could have demonstrated the presence of organic matter (carbon) over the MNP. But there is no reference to this.

A/R: Thank you for this valuable comment and suggestion. The authors have now removed this sentence from the revised manuscript as suggested by the reviewer.

  1. Result and discussion. This section should not make reference to magnetic study, because thereis no other subsection… Besides a great need of rephrasing the whole text, there is a continued lack of references when discussing key questions. For the sake of a better understanding, I will use an example: in lines 176-186, besides the need of rephrasing, the authors claim something which is not evident for all readers, so one or more references in different claims should be used to justify their reasonings. Same stands for the rest of the section.

Response: Dear reviewer thank you very much for the valuable and remarkable suggestion, now some more references have been added in the manuscript.

  1. Conclusion. Some rephrasing should be performed in order to improve this key section.

 In conclusion, there is a lot of effort to be put into this manuscript in order to improve it significantly.

However, the experimental part seems to be correct, therefore I recommend accepting the manuscript after a minor revision (though the text editing should be quite extensive).

A/R: Thank you for this valuable comment and suggestion. The authors have now modified the conclusion section as per the suggestion of the revised in the revised manuscript.

Reviewer 3 Report

In this manuscript entitled "Green synthesis of Unsaturated Fatty acid Mediated Magnetite Nanoparticles and its Structural & Magnetic Studies", the authors have studied the structure and magnetic properties of magnetite nanoparticles. The research on structure and magnetism is rich and systematic. However, a large number of data images are blurry or even completely invisible, thus, this paper should be reconsidered after necessary revisions.

1. Fig. 4, Fig. 5 and Fig. 6 are too vague, please provide clear data.

2. The particle size distribution in SEM pictures should be provided.

3. Some very recently updated articles [such as Hui Zhang, Yan Wang, Haiou Wang, Dexuan Huo, and Weishi Tan, Journal of Applied Physics 131, 043901 (2022)] related to structure, surface morphology (particle size distribution) and magnetic properties of magnetic materials are suggested to be cited in the introduction or discussion part.

4. The EDS should be provided.

5. Magnetic data (Fig. 7 and 8) are completely invisible, so it is impossible to make effective comments on related contents.

Author Response

In this manuscript entitled "Green synthesis of Unsaturated Fatty acid Mediated Magnetite Nanoparticles and its Structural & Magnetic Studies", the authors have studied the structure and magnetic properties of magnetite nanoparticles. The research on structure and magnetism is rich and systematic. However, a large number of data images are blurry or even completely invisible, thus, this paper should be reconsidered after necessary revisions.

  1. 4, Fig. 5 and Fig. 6 are too vague, please provide clear data.

Response: Dear reviewer thank you very much for the valuable suggestion, now we have added new with the high-resolution figure in the manuscript.

  1. The particle size distribution in SEM pictures should be provided.

Response: Dear reviewer thank you for the valuable suggestion, now we have provided particle size distribution in the manuscript.

  1. Some very recently updated articles [such as Hui Zhang, Yan Wang, Haiou Wang, Dexuan Huo, and Weishi Tan, Journal of Applied Physics 131, 043901 (2022)] related to structure, surface morphology (particle size distribution) and magnetic properties of magnetic materials are suggested to be cited in the introduction or discussion part.

Response: Dear reviewer thank you for the nice suggestions now we have cited this article in the manuscript.

  1. The EDS should be provided.

Response: Dear reviewer thank you very much for the valuable comment, we didn’t study EDS but in the next study we will discuss definitely it.

  1. Magnetic data (Fig. 7 and 8) are completely invisible, so it is impossible to make effective comments on related contents.

Response: Dear reviewer thank you for the valuable suggestions, now we have added both the new figures in the manuscript.

Round 2

Reviewer 3 Report

This paper has been improved and can be accepted.

Author Response

Thank you for accepting the manuscript.